# Gut Microbiota Profile and Changes in Body Weight in Elderly Subjects with Overweight/Obesity and Metabolic Syndrome

**DOI:** 10.3390/microorganisms9020346

**Published:** 2021-02-10

**Authors:** Alessandro Atzeni, Serena Galié, Jananee Muralidharan, Nancy Babio, Francisco José Tinahones, Jesús Vioque, Dolores Corella, Olga Castañer, Josep Vidal, Isabel Moreno-Indias, Laura Torres-Collado, Rebeca Fernández-Carrión, Montserrat Fitó, Romina Olbeyra, Miguel Angel Martínez-González, Monica Bulló, Jordi Salas-Salvadó

**Affiliations:** 1Department of Biochemistry and Biotechnology, Universitat Rovira i Virgili, 43201 Reus, Spain; alessandro.atzeni@urv.cat (A.A.); serena.galie@iispv.cat (S.G.); jananee.muralidharan@urv.cat (J.M.); nancy.babio@urv.cat (N.B.); 2Institut D’Investigació Sanitària Pere Virgili (IISPV), Hospital Universitari de Sant Joan de Reus, 43204 Reus, Spain; 3CIBER de Fisiopatología de la Obesidad y la Nutrición (CIBEROBN), Instituto de Salud Carlos III, 28029 Madrid, Spain; fjtinahones@uma.es (F.J.T.); dolores.corella@uv.es (D.C.); ocastaner@imim.es (O.C.); isabel.moreno@ibima.eu (I.M.-I.); rebeca.fernandez@uv.es (R.F.-C.); mfito@imim.es (M.F.); mamartinez@unav.es (M.A.M.-G.); 4Unidad de Gestión Clínica de Endocrinología y Nutrición, Instituto de Investigación Biomédica de Málaga (IBIMA), Hospital Universitario Virgen de la Victoria, 29010 Málaga, Spain; 5Instituto de Investigación Sanitaria y Biomédica de Alicante, ISABIAL-UMH, 03010 Alicante, Spain; vioque@umh.es (J.V.); l.torres@umh.es (L.T.-C.); 6CIBER Epidemiología y Salud Pública (CIBERESP), Instituto de Salud Carlos III (ISCIII), 28029 Madrid, Spain; 7Department of Preventive Medicine, University of Valencia, 46100 Valencia, Spain; 8Cardiovascular Risk and Nutrition (Regicor Study Group), Hospital del Mar Research Institute (IMIM), 08003 Barcelona, Spain; 9Endocrinology and Nutrition Department, Institut d’Investigacions Biomèdiques August Pi Sunyer (IDIBAPS), Hospital Clinic Universitari, 08036 Barcelona, Spain; jovidal@clinic.cat (J.V.); rominaolbeyra@gmail.com (R.O.); 10CIBER Diabetes y Enfermedades Metabólicas (CIBERDEM), Instituto de Salud Carlos III (ISCIII), 28029 Madrid, Spain; 11Department of Preventive Medicine and Public Health, University of Navarra, IdiSNA, 31008 Pamplona, Spain; 12Department of Nutrition, Harvard T.H. Chan School of Public Health, Harvard University, Boston, MA 02115, USA

**Keywords:** obesity, gut microbiota, BMI, weight loss, 16S sequencing, clinical trial

## Abstract

Gut microbiota is essential for the development of obesity and related comorbidities. However, studies describing the association between specific bacteria and obesity or weight loss reported discordant results. The present observational study, conducted within the frame of the PREDIMED-Plus clinical trial, aims to assess the association between fecal microbiota, body composition and weight loss, in response to a 12-month lifestyle intervention in a subsample of 372 individuals (age 55–75) with overweight/obesity and metabolic syndrome. Participants were stratified by tertiles of baseline body mass index (BMI) and changes in body weight after 12-month intervention. General assessments, anthropometry and biochemical measurements, and stool samples were collected. 16S amplicon sequencing was performed on bacterial DNA extracted from stool samples and microbiota analyzed. Differential abundance analysis showed an enrichment of *Prevotella 9*, *Lachnospiraceae* UCG-001 and *Bacteroides*, associated with a higher weight loss after 12-month of follow-up, whereas in the cross-sectional analysis, *Prevotella 2* and *Bacteroides* were enriched in the lowest tertile of baseline BMI. Our findings suggest that fecal microbiota plays an important role in the control of body weight, supporting specific genera as potential target in personalized nutrition for obesity management. A more in-depth taxonomic identification method and the need of metabolic information encourages to further investigation.

## 1. Introduction

Overweight and obesity are considered a worldwide public health problem which has rapidly increased up to reach global epidemic proportions [1]. Obesity is a complex multifactorial disease, characterized by an anomalous or disproportionate adipose tissue accumulation associated with several metabolic complications [2,3].

In the last few years, gut microbiota has been highlighted as an important factor related to obesity and its associated comorbidities [4]. Causal evidence linking gut microbiota to obesity mostly originates from fecal transplant studies conducted in germ free mice that gained weight when colonized with gut microbes from obese donors [5]. Moreover, the gut microbiota is able to predict post-dieting weight regain in obese mice [6]. A recent systematic review of observational studies has reported differences between the gut microbiota profiles of individuals with obesity and lean individuals, identifying some bacteria potentially involved in the development of obesity [4]. *Bacteroidetes* are commonly less abundant in people with obesity, with this abundance increasing along with weight-loss [7], whereas *Firmicutes* phylum as some of their genera as *Lactobacillus* and *Clostridium* have been associated to metabolic dysregulations related to obesity [8], suggesting that specific bacteria could be beneficial or detrimental to obesity. Whether the gut microbes are related to weight dynamics in humans has been sparsely studied [9]. In a weight-loss study conducted over 49 participants from the DIETFITS randomized either to a low-carbohydrates or low-fat diets, microbiota composition did not predict participants’ weight loss at 1 year [10]. In contrast, other trials of shorter duration shown that different relative abundance of specific genera (i.e., *Phascolarctobacterium*, *Dialister*, *Prevotella*-to-*Bacteroidetes* ratio) were associated with a higher or lower weight loss [11,12].

Accordingly, the aim of the present study is to identify, in a large sample size, specific genera associated with baseline body mass index (BMI) and changes in body weight in response to a lifestyle intervention, in an elderly population with overweight/obesity and metabolic syndrome.

## 2. Materials and Methods

### 2.1. Participants and Study Design

This study was conducted within the frame of the PREDIMED-Plus clinical trial, that aims to evaluate the long-term effect of an intensive weight-loss lifestyle intervention on cardiovascular disease and mortality in a population with overweight and obesity (BMI 27–40 kg/m^2^), aged between 55 and 75 years old and who at least met 3 criteria for metabolic syndrome [13]. Participants were randomized in a 1:1 ratio to an intervention group that encouraged an energy-reduced Mediterranean diet, promoted physical activity, and provided behavioral support, or to a control group that encouraged an energy-unrestricted Mediterranean diet without any other specific advice for losing weight. The PREDIMED-Plus study protocol is available at http://www.predimedplus.com, accessed 18 November 2020, and was registered at the International Standard Randomized Controlled Trial (http://www.isrctn.com/ISRCTN89898870, accessed 18 November 2020). This trial was approved by the institutional review board of all participating institutions, and participants provided written informed consent.

The present observational study included 400 participants (200 participants for each intervention group) recruited in the PREDIMED-Plus centers of Reus and Málaga in Spain, randomly selected, matched by sex, age, and BMI, and with stool samples available at baseline and after 12-month of intervention.

A cross-sectional analysis was conducted stratifying the sample by tertiles of baseline BMI. In addition, a longitudinal analysis was conducted stratifying the sample by tertiles of changes in body weight after 12-month intervention.

### 2.2. General Assessments, Anthropometric and Biochemical Measurements, Samples Collection

Information about disease prevalence, lifestyle and medication use was collected. At baseline and 12-month timepoint, waist circumference was measured midway between the the lowest rib and the iliac crest using an anthropometric tape, body weight was measured using high-quality electronic calibrated scales, height was measured using a wall-mounted stadiometer. Systolic and diastolic blood pressure was measured 3 times using a validated semiautomatic oscillometer (Omron HEM-705CP, Kyoto, Japan) and the mean value recorded.

Blood samples were collected at both timepoints after an overnight fast. Plasma fasting glucose, total cholesterol, high-density lipoprotein (HDL) cholesterol, and triglycerides concentrations were measured using standard enzymatic methods, low-density lipoprotein (LDL) cholesterol concentrations were calculated with the Friedewald formula whenever triglycerides were less than 300 mg/dL, and glycated hemoglobin was measured by a chromatographic method.

Baseline and 12-month timepoint stool samples were collected and kept frozen till the delivery to the laboratory. In case of antibiotic treatment or fiber supplements, samples were collected 15 days after treatment completion. Stool samples were then separated into 250 mg aliquots stored at −80 °C, until analysis.

### 2.3. Microbial DNA Extraction, 16S Amplicon Sequencing and Data Processing

Microbial DNA was extracted using the QIAmp PowerFecal DNA kit (Qiagen, Hilden, Germany) following the manufacturer’s instructions. In the first step of the extraction, an additional lysing of 5 min using FastPrep-24™ 5G Homogenizer (MP Biomedicals, Santa Ana, CA, USA) was conducted. DNA concentration and purity were assessed with the Qubit 2.0 Fluorometer-dsDNA Broad Range Assay Kit (Invitrogen, Carlsbad, CA, USA).

Targeted sequencing libraries were created with the 16S Metagenomics kit (Life Technologies, Carlsbad, CA, USA), using a pool of primers to amplify multiple hypervariable regions (V2, V3, V4, V6-7, V8, V9) of the 16S rRNA gene, in combination with Ion Plus Fragment Library Kit (Life Technologies, Carlsbad, CA, USA), to ligate barcoded adapters. Synthesized libraries were pooled and templated on the automated Ion Chef system (Life Technologies, Carlsbad, CA, USA) followed by a 400 bp sequencing on the Ion S5 (Life Technologies, Carlsbad, CA, USA). Sequenced reads were generated in BAM (Binary Alignment Map) format and then converted in FASTQ format using the File Explorer plugin of the Torrent Suite Server software (Life Technologies, Carlsbad, CA, USA), interfaced with the Ion S5.

A customized Python script [14] was used to separate the reads according to the different hypervariable regions of the 16S rRNA gene, and the V4 data selected and individually processed with the software QIIME (Quantitative Insight into Microbial Ecology) 2, version 2020.2 [15]. Sequenced reads were demultiplexed, trimmed to 265 bp, and denoised into ASVs (amplicon sequence variants) using the denoise-pyro method of the DADA2 plugin [16]. Taxonomic assignment was performed using the consensus-vsearch method of the vsearch plugin [17], against the 16S rRNA gene reference database SILVA 132 [18].

### 2.4. Statistical Analysis

Baseline characteristics of participant were described as means and standard deviations or median and interquartile range (as appropriate) for quantitative variables, and numbers and percentages for categorical variables. Population was stratified by tertiles of baseline BMI and by tertiles of changes in body weight after 12-month intervention irrespective of the intervention group of the trial. Differences across tertiles were evaluated through one-way analysis of variance (ANOVA) or Kruskal–Wallis test for numerical variables, as appropriate, and with Pearson’s chi-square test for categorical variables. Student’s t-test or Mann–Whitney U test were used to calculate differences between tertiles for numerical variables, Pearson’s chi-square test was used for categorical variables. Statistical analysis was carried out using IBM SPSS Statistics version 23 (SPSS Inc., Chicago, IL, USA). All statistical tests were 2-sided and *P* value < 0.05 was deemed statistically significant.

ASV counts and taxonomic information generated with QIIME 2, were imported into R (version 3.6.2) and processed with the package Phyloseq, version 1.30.0 [19]. ASVs counts table was filtered at 10% prevalence cut off at genus level for both samples and overall ASVs.

Chao1, Shannon and Simpson indexes were calculated and pairwise comparison using Wilcoxon rank sum test performed to evaluate differences in microbial diversity among tertiles of baseline BMI. Bray-Curtis, Jaccard, Weighted and Unweighted Unifrac distance matrices were calculated and permutational multivariate analysis of variance (PERMANOVA) performed using the adonis function (“vegan” package, version 2.5-6), to test differences in groups compositions, whereas permutation test for homogeneity of multivariate dispersions was performed to test variability among groups.

The log-normalized *Firmicutes*-to-*Bacteroidetes* (F/B) ratio was computed based on the relative abundance between the phylum *Firmicutes* and *Bacteroidetes*, the log-normalized *Prevotella*-to-*Bacteroides* ratio (P/B) was computed based on the relative abundance between the genus *Prevotella* and *Bacteroides*. One-way ANOVA was used to test if F/B and P/B ratios were statistically significant different between tertiles of baseline BMI and tertiles of changes in body weight after 12-month intervention.

Differential abundant significant ASVs (Benjamini–Hochberg adjusted *P* value < 0.05) were identified between tertiles of baseline BMI and tertiles of changes in body weight after 12-month intervention, using Wald’s test in the DESeq2 package, version 1.26.0 [20], adjusting for type 2 diabetes prevalence and intervention group as covariates.

## 3. Results

### 3.1. Association between Fecal Microbiota and Tertiles of Baseline Body Mass Index

A total of 400 participants, in the framework of the PREDIMED-Plus clinical trial, were randomly selected and matched by age, sex and BMI. From these 400, stool samples at baseline and at 12-month timepoint were available for 372, from which bacterial DNA was extracted and sequenced. Sequence data generated was separated according to the different hypervariable regions of the 16S rRNA gene, and V4 data selected and processed with QIIME 2. Few samples were excluded from the analysis because no information was generated after the denoise step, or because missed or repeated, reducing the number of participants included in the cross-sectional study to 368. Finally, counts table was filtered at 10% prevalence cut off at genus level for both samples and overall ASVs, further reducing the number of participants to 364.

The baseline characteristics of the study population categorized by tertiles of baseline BMI are shown in Table 1. Body weight, BMI, waist circumference, fasting glucose and glycated hemoglobin levels, the prevalence of type 2 diabetes, and the prevalence of metformin or other antidiabetic drugs use, were higher in the tertiles 2 and 3 compared to tertile 1.

Differences in alpha and beta diversity, as well as differences in F/B ratio and P/B ratio between tertiles were not statistically significant (Appendix A).

A total of 5453 ASVs were detected in 364 samples. Statistically significant differential abundant ASVs between tertiles of baseline BMI are summarized in Figure 1, whereas detailed information, including *P* values are listed in Appendix A. The analysis revealed one ASV representing the genus *Prevotella* 2, more abundant in tertile 1 versus to tertile 2, one ASV representing the genus *Bacteroides* in tertile 1 versus tertile 3, one ASV representing *Bacteroides* in tertile 2 versus tertile 3 and one ASV representing the genus *Prevotella* 2 in tertile 3 versus tertile 2.

### 3.2. Association between Fecal Microbiota and Tertiles of Changes in Body Weight after 12-Month Intervention

From 372 participants with available stool samples, 357 were those with available baseline and correspondent 12-month timepoint sample included in the following steps of the analysis. Following the counts table filtering step, 12 samples were excluded from the analysis, further reducing the number of samples to 345.

Baseline characteristics and changes at 12-month timepoint in anthropometric and biochemical parameters, and blood pressure are shown in Table 2. In average, participants in tertile 1 and 2 lose weight (−7.2 ± 3.4 kg and −2.3 ± 1.0 kg, respectively), whereas participants in tertile 3 increased weight during the intervention. A total 82.6%, 54.8% and 13.0% of subjects allocated in tertiles 1, 2 and 3, respectively, belonged to the intensive lifestyle intervention group. There were significant differences at baseline in BMI, waist circumference and glucose levels across tertiles. Glucose levels were higher in those participants in tertile 2 compared to those in the other tertiles. Total body weight, BMI, waist circumference, glucose levels, glycated hemoglobin, and diastolic blood pressure decreased in tertile 1 and increased in tertile 3, with differences in changes significant between both extreme tertiles of body weight changes.

Differences in F/B ratio and P/B ratio were not statistically significant across tertiles of changes in body weight (Appendix A).

A total of 8060 ASVs were detected in 690 samples. Statistically significant differential abundant ASVs determined between tertiles of changes in body weight after 12-month intervention are summarized in Figure 2, whereas detailed information, including *P* values are listed in Appendix A. A total of six ASVs were differentially abundant between tertile 1 and tertile 2, of which five (mostly represented by genera Prevotella 9, Bacteroides, and Lachnospiraceae UCG-001) were more abundant in tertile 1, whereas one ASV (represented by Prevotella 2 genus) more abundant in tertile 2. A total of six ASVs were differentially abundant between tertile 1 and tertile 3, all of which (mostly represented by genera Prevotella 9, Lachnospiraceae UCG-001, Bacteroides and uncultured bacteria) were more abundant in tertile 1. A total of 18 ASVs were differentially abundant between tertile 2 and tertile 3, of which two (represented by Bacteroides and Prevotella 2 genus) were more abundant in tertile 2, and 16 (mostly represented by genera Sutterella, Bacteroides, Prevotella 2, Dialister, Prevotella 9) were more abundant in tertile 2.

## 4. Discussion

In our study, conducted on subjects with obesity/overweight and metabolic syndrome, we found that a significant differential abundance of ASVs representing *Prevotella* 9, *Lachnospiraceae* UCG-001 and *Bacteroides* genus, was associated with a higher weight loss after 12-month of follow-up. Our findings support the hypothesis that specific components of fecal microbiota may be involved in the control of body weight. Consistently, in the cross-sectional analysis, ASVs representing *Prevotella* 2 and *Bacteroides* genus were significantly differentially abundant in the lowest tertile of baseline BMI.

The role of gut microbiota in the control of body weight was first described by Bäckhed et al. which observed an increase of body fat content and insulin resistance in GF mice colonized with gut microbiota of conventionally raised mice [7]. A drastic reduction in *Bacteroidetes* and a proportional increase in *Firmicutes* was described in genetically obese mice compared to lean and wild type animals fed with the same diet, highlighting the gut microbiota’s contribution to obesity [21]. Further animal [22] and human studies [7] confirmed these results; however, these findings are not consistent across different studies. A study conducted by Duncan et al., with the objective to examine the associations betweenBMI, weight loss and fecal microbiota, showed no significant differences in the proportion of *Bacteroidetes* between individuals with obesity and healthy individuals [23], whereas other studies, described a higher relative abundance of *Bacteroidetes* in subjects with obesity compared with lean subjects [24,25]. Accordingly, we did not find any association between F/B ratio neither with baseline BMI nor with weight changes, highlighting the need for focusing on a deeper taxonomic level rather than just consider the imbalance in the proportion of *Bacteroidetes* and *Firmicutes* phylum [26].

Studies at genus level showed that *Bacteroides* were lower in individuals with obesity compared to healthy individuals [27]. In a study conducted by Liu et al. *Bacteroides* spp. was found markedly reduced in Chinese individuals with obesity [28]. On the other hand, a comparative analysis of the gut microbiota of lean, normal, individuals with obesity and surgically treated Indian individuals with obesity showed higher levels of *Bacteroides* among subjects with obesity and its abundance positively correlated with BMI [25]. On the contrary, in our study, the *Bacteroides* abundance was significantly enriched in those patients with lower baseline BMI, and those who lost weight after 12-month of lifestyle intervention. *Bacteroides* is known as a mutualist bacterium that could drive the functionality of others [29]. Moreover, *Bacteroides* is able to adapt its metabolic machinery to the food source [30], becoming a key bacterium for dietary and/or weight loss interventions.

The relative abundance of *Prevotella* was found increased in individuals with severe obesity [31], contrarily in our results *Prevotella* genus was found increased in the lowest tertile of baseline BMI and in highest tertile of weight loss. Even though, other studies did not show any correlation between increased abundance of *Prevotella* and BMI [32].

The P/B ratio was demonstrated to be a useful tool to evaluate weight loss success in individuals with obesity exposed to ad libitum high fiber diets [12]. Results showed that individuals with high P/B ratio were more susceptible to lose weight on a diet rich in fiber and whole grains. A more recent study aimed to investigate the differences in weight loss maintenance between subjects with low and high P/B ratio and the potential interactions with markers of glucose metabolism and dietary fiber intake. Results showed that subjects with high P/B ratio were more susceptible to regain body weight than subjects with low P/B ratio, especially when dietary fiber intake was low and glucose metabolism was impaired [33]. Considering these findings, matching diet to gut microbiota profile may be crucial to increasing the effectiveness of weight loss programs. In a recent study conducted by Christiansen et al., healthy overweight subjects exposed to different fiber-rich diet were stratified according to baseline P/B ratios and *Prevotella* abundance. The *Prevotella* abundances correlated inversely with weight changes, whereas P/B ratios did not show any correlation. Subjects with high *Prevotella* abundance lost more weight than subjects with low *Prevotella* abundance when consuming a fiber-rich diet [34]. These outcomes are only partly supported by our results, in which no significant differences were observed in P/B ratio, but *Prevotella* 9 genus was found to be increased after weight loss.

Changes in the gut microbiota of patients with obesity after weight-loss interventions, have been described with divergent results between studies in terms of the bacterial profile involved [7,35]. A study conducted by Korpela et al. presented evidence about the validity of the baseline microbiota information in predicting the host’s response to a dietary intervention [36]. Specifically, they identified *Clostridium* clusters and *Bacilli* indicative of the amenability of the gut microbiota to dietary modification, which in turn was associated with the host’s lipid metabolism. According to these findings, we also detected an enrichment of uncultured genera belonging to the *Clostridiales* order in those patients with more tendency to lose weight after lifestyle intervention.

Contrary to our expectations in our study we observed that *Lachnospiraceae* UCG-001 genus was more abundant at baseline, in those subjects who lose weight after 12-month intervention, as these genera are producers of short chain fatty acids involved in an improvement in energy efficiency [37]. However, in a recent review, inconsistencies across different studies, about the impact of *Lachnospiaraceae* on the energy efficiency were reported [37], probably because an adequate amount of short chain fatty acids is necessary to control energy intake and expenditure.

A recent review showed inconsistent evidence to support baseline gut microbiota as an accurate predictor of weight loss in obesity, suggesting the need of further investigation with larger scale [38]. A recent study by Fragiadakis et al., aimed to determine if baseline gut microbiota was associated with long-term (12-month) diet weight loss success [39]. After 3 months of weight loss, they show differences in gut microbiota profile, however gut bacteria returned to the original composition at 12 months. Baseline gut microbiota profile was not associated to long-term changes in total body weight, suggesting a resilience to perturbation of the microbiota starting profile. Contrary to the aforementioned study, we have been able to detect differences at genus level after 12-month intervention, supporting long-term effects on weight loss.

In addition to the large sample size and the homogeneity of our study population (all with overweight/obesity and metabolic syndrome), this study has some limitations that deserve comments. First, in our study we did not evaluate short-term changes in body weight and therefore, we cannot determine resilience of the gut microbiota; second, the design of our study did not allow it to establish causality; and finally, as this study was conducted in elderly Spanish individuals with obesity and metabolic syndrome, it cannot be extrapolated to other populations.

## 5. Conclusions

We identified specific fecal microbiota signatures at genus level potentially related to changes in body weight in response to lifestyle intervention in an elderly population with overweight and obesity. These finding offer a promising novel perspective to support clinicians to tailor personalized interventions for obesity treatment, in which successful strategies can be predicted according to the microbiota composition. In any case, the validity of these microbial signatures has to be reproduced in other populations, taking into account the gut microbiota at species level. Furthermore, metabolic data are necessary to integrate these results and identify potential pathways involved, encouraging the need for further investigation in this field.

## Figures and Tables

**Figure 1 microorganisms-09-00346-f001:**
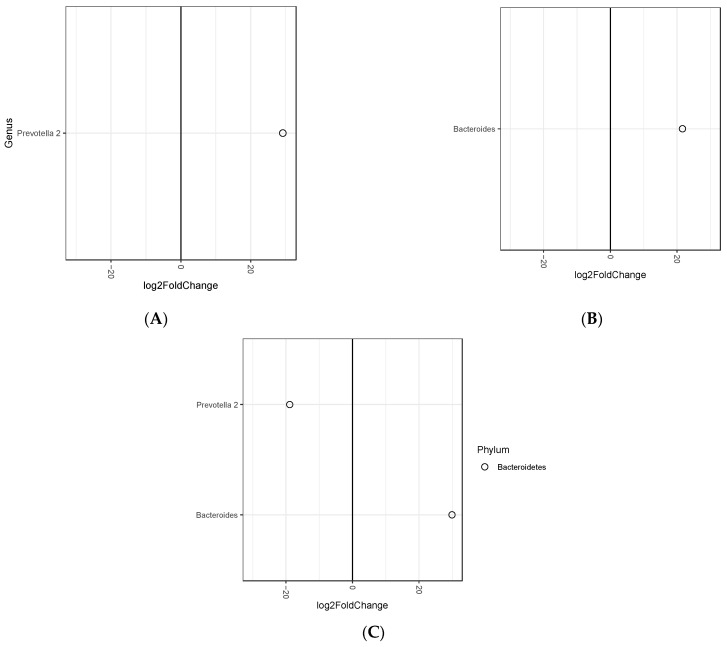
Differential abundant ASVs between tertiles of baseline body mass index. (**A**) tertile 1 versus tertile 2, (**B**) tertile 1 versus tertile 3, (**C**) tertile 2 versus tertile 3. Only ASVs with adjusted *P*-values < 0.05 are depicted.

**Figure 2 microorganisms-09-00346-f002:**
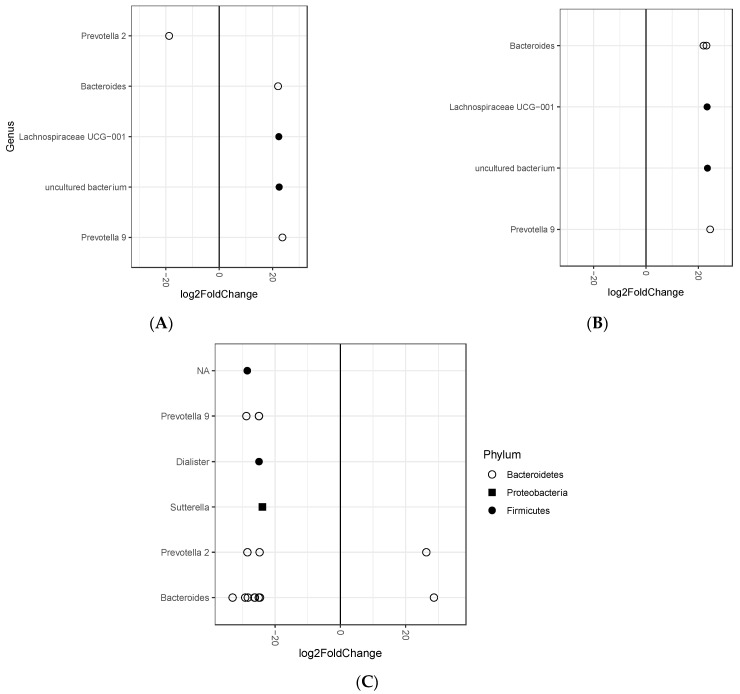
Differential abundant ASVs between tertiles of changes in body weight after 12-month intervention. (**A**) tertile 1 versus tertile 2, (**B**) tertile 1 versus tertile 3, (**C**) tertile 2 versus tertile 3. Only ASVs with adjusted *P*-values < 0.05 are depicted.

**Table 1 microorganisms-09-00346-t001:** Baseline characteristics of the study population according to tertiles of baseline body mass index.

Tertile Min–Max	T1 (*n* = 121) 25.9–31.5	T2 (*n* = 122) 31.5–35.0	T3 (*n* = 121) 35.0–40.3	*P* Trend ^&^
Sex, female	58 (47.9)	57 (46.7)	73 (60.3)	0.064
Age, years	64.9 ± 5.2	64.3 ± 4.8	65.0 ± 5.1	0.591
Intervention group	55 (45.5)	63 (51.6)	66 (54.5)	0.352
Body weight, kg	79.4 ± 9.1	88.4 ± 10.4 **	96.9 ± 12.0 **†	<0.001
BMI, kg/m^2^	29.4 ± 1.4	33.1 ± 1.0 **	37.3 ± 1.5 **†	<0.001
Waist circumference, cm	102.2 ± 7.1	109.7 ± 7.4 **	117.5 ± 8.1 **†	<0.001
Smoking Current smoker Former smoker Never smoked	20 (16.5) 48 (39.7) 52 (43.0)	21 (17.2) 47 (38.5) 54 (44.3)	15 (12.4) 39 (32.2) 67 (55.4)	0.369
Education Primary school Secondary school Academic or graduate	64 (52.9) 37 (30.6) 20 (16.5)	68 (55.7) 39 (32.0) 15 (12.3)	64 (52.9) 41 (33.9) 16 (13.2)	0.880
Recruiting center				0.093
Reus	45 (37.2)	39 (32.0)	55 (45.5)	
Malaga	76 (62.8)	83 (68.0)	66 (54.5)	
Hypercholesterolemia	77 (63.6)	82 (67.2)	75 (62.0)	0.685
Hypertension	110 (90.9)	117 (95.9)	116 (95.9)	0.159
T2DM prevalence	17 (14.0)	33 (27.0) *	35 (28.9) *	0.012
Insulin treatment	2 (1.7)	9 (7.4)	10 (8.3)	0.057
Metformin treatment	10 (8.3)	29 (23.8) *	26 (21.5) *	0.003
Other anti diabetic drugs use	12 (9.9)	27 (22.1) *	28 (23.1) *	0.013
Glucose, mg/dL	103.9 ± 19.8	112.4 ± 28.7 *	112.9 ± 25.8 *	0.007
HbA1c, %	5.7 [0.6]	5.9 [0.6] **	5.9 [0.8] *	0.001
Triglycerides, mg/dL	152 [100]	147 [90]	155.5 [78]	0.291
Total cholesterol, mg/dL	204.8 ± 38.6	197.0 ± 37.2	203.0 ± 37.1	0.241
HDL-cholesterol, mg/dL	50.3 ± 12.9	48.0 ± 12.5	48.6 ± 11.9	0.316
LDL-cholesterol, mg/dL	122.6 ± 34.2	114.8 ± 33.1	118.8 ± 33.0	0.193
SBP, mm Hg	139.0 ± 18.2	140.2 ± 14.8	141.3 ± 17.6	0.589
DBP, mm Hg	78.8 ± 9.6	80.6 ± 9.6	77.9 ± 10.5	0.099

Data shown as mean ± SD, median [IQR] or *n* (%); SD; standard deviation; IQR; interquartile range; BMI, body mass index; T2D, type 2 diabetes; HbA1c, glycated hemoglobin; HDL, high-density lipoprotein; LDL, low-density lipoprotein SBP, systolic blood pressure; DBP, diastolic blood pressure. ^&^ One-way ANOVA, Pearson’s chi-square test or Kruskal–Wallis test used to calculate differences across tertiles; Pearson’s chi-square test, Student’s t-test or Mann–Whitney test used to calculate differences between tertiles; ** *P* < 0.001 vs. T1; * *P* < 0.05 vs. T1; † *P* < 0.001 vs. T2.

**Table 2 microorganisms-09-00346-t002:** Baseline characteristics and changes of the study population according to tertiles of changes in body weight after 12-month intervention.

Tertile Min—Max	T1 (*n* = 115) −24.2–−4.5	T2 (*n* = 115) −4.5–−0.7	T3 (*n* = 115) −0.72–11.6	*P* Trend ^&^
Sex, female	54 (47.0)	62 (53.9)	57 (49.6)	0.567
Age, years	64.4 ± 5.1	64.8 ± 4.8	64.8 ± 5.3	0.788
Recruiting center				0.178
Reus	45 (39.1)	48 (41.7)	35 (30.4)	
Malaga	70 (60.9)	67 (58.3)	80 (69.6)	
Intervention group	95 (82.6)	63 (54.8) **	15 (13.0) **	<0.001
Hypercholesterolemia	69 (60.0)	72 (62.6)	78 (68.7)	0.455
Hypertension	105 (91.3)	110 (95.7)	109 (94.8)	0.345
Type 2 diabetes prevalence	25 (21.7)	35 (30.4)	21 (18.3)	0.081
Insulin treatment	4 (3.5)	10 (8.7)	6 (5.2)	0.226
Metformin treatment	19 (16.5)	26 (22.6)	17 (18.3)	0.268
Other anti diabetic drugs use	19 (16.5)	28 (24.3)	17 (14.8)	0.139
Body weight, kg	89.2 ± 13.0	89.6 ± 14.1	86.1 ± 10.9	0.066
Change in body weight, kg	−7.2 ± 3.4	−2.3 ± 1.0 **	1.5 ± 1.7 **††	<0.001
BMI, kg/m^2^	33.3 ± 3.6	33.8 ± 3.6	32.5 (3.1) †	0.018
Change in BMI, kg/m^2^	−2.6 ± 1.3	−0.8 ± 0.5 **	0.6 ± 0.7 **††	<0.001
Waist circumference, cm	110.4 ± 10.1	111.6 ± 10.2	107.6 ± 9.8 *†	0.007
Change in waist circumference, cm	−7.4 ± 4.7	−2.4 ± 3.8 **	1.2 ± 3.8 **††	<0.001
Glucose, mg/dL	107.9 ± 22.7	114.9 ± 30.5 *	106.3 ± 21.3 †	0.023
Change in glucose, mg/dL	−7.8 ± 15.8	−1.9 ± 21.1 *	3.6 ± 21.4 **†	<0.001
HbA1c, %	5.9 [0.6]	5.9 [0.9]	5.7 [0.6]	0.086
Changes in HbA1c, %	−0.2 [0.4]	0.0 [0.3] *	0.1 [0.3] **†	<0.001
Triglycerides, mg/dL	137.0 [78.0]	153.0 [98.0]	162.0 [92.0]	0.571
Change in triglycerides, mg/dL	−19.0 [60.0]	−8.5 [60.2]	−4.5 [76.2]	0.595
Total cholesterol, mg/dL	201.5 ± 31.1	195.5 ± 40.2	205.1 ± 40.9	0.169
Change in total cholesterol, mg/dL	−1.6 ± 27.3	−0.8 ± 31.7	−4.9 ± 39.2	0.614
HDL-cholesterol, mg/dL	48.2 ± 13.3	48.0 ± 12.0	50.1 ± 12.2	0.387
Change in HDL-cholesterol, mg/dL	3.0 ± 6.7	3.0 ± 7.2	1.0 ± 8.2	0.065
LDL-cholesterol, mg/dL	120.4 ± 28.7	113.9 ± 34.5	120.7 ± 36.9	0.232
Change in LDL-cholesterol, mg/dL	−1.2 ± 24.2	−0.8 ± 27.1	−5.2 ± 35.3	0.462
SBP, mm Hg	140.1 ± 15.6	141.7 ± 17.3	139.2 ± 17.3	0.531
Change in SBP, mm Hg	−6.8 ± 13.1	−4.1 ± 16.0	−2.0 ± 16.6	0.058
DBP, mm Hg	79.9 ± 9.6	79.0 ± 10.2	79.0 ± 10.1	0.713
Change in DBP, mm Hg	−3.6 ± 8.2	−1.0 ± 8.3 *	−1.1 ± 8.3 *	0.027

Data shown as mean ± SD, median [IQR] or *n* (%); SD; standard deviation; IQR; interquartile range; BMI, body mass index; T2D, type 2 diabetes; HbA1c, glycated hemoglobin; HDL, high-density lipoprotein; LDL, low-density lipoprotein SBP, systolic blood pressure; DBP, diastolic blood pressure. ^&^ One-way ANOVA, Pearson’s chi-square test or Kruskal–Wallis test used to calculate differences across tertiles; Pearson’s chi-square test, Student’s t-test or Man-Whitney test used to calculate differences between tertiles; ** *P* < 0.001 vs. T1; * *P* < 0.05 vs. T1; †† *P* < 0.001 vs. T2; † *P* < 0.05 vs. T2.

## Data Availability

The datasets generated and analysed during the current study are not publicly available due to data regulations and for ethical reasons, considering that this information might compromise research participants’ consent because our participants only gave their consent for the use of their data by the original team of investigators. However, collaboration for data analyses can be requested by sending a letter to the PREDIMED-Plus steering Committee (predimed_plus_scommittee@googlegroups.com). The request will then be passed to all the members of the PREDIMED-Plus Steering Committee for deliberation.

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
