# Peer review of "Gut Microbiota Profile and Changes in Body Weight in Elderly Subjects with Overweight/Obesity and Metabolic Syndrome"

_microorganisms, 2021, doi:10.3390/microorganisms9020346_

Round 1
Reviewer 1 Report
Easy to read. Analysis as much as can be expected given the results.The interplay is interesting although not surprising.The authors give an objective critic of their own study.
Author Response
We greatly appreciate the reviewer overall appreciation of our m/s, and thank for their comments and suggestions that have improved the final version of our m/s.
Reviewer 2 Report
While this is an interesting study, it has small limitations that make meaningful conclusions difficult.
Specific comments:
1. It was shown, that antidiabetic drugs show positive effects on dysbiosis; generally without any consensus specifically regarding the Firmicutes to Bacteroidetes ratio. Thus, beneficial effects might be mediated by specific taxa. However, Metformin act din different fashion. In addition to its glucose-lowering and insulin sensitizing effects, metformin promotes SCFA-producing and mucin-degrading bacteria. Authors should briefly discuss how Metformin (and other antidiabetic drugs) could intefere with their results. Authors could also cite works by Montandon SA (Genes), Whang (Lancet) and Wu (Nature).
2. It will be interesting to know the spectrum cultures in obese who lost weight after the intervntion dependiing of antidiabetic drug administered.
3.The microbiome spectrum in any study population will also depend on the regional prevalence. That information (comparing described results with other authors) is at least important as other comparisons.
Author Response
While this is an interesting study, it has small limitations that make meaningful conclusions difficult.
Specific comments:
- It was shown that antidiabetic drugs show positive effects on dysbiosis; generally, without any consensus specifically regarding the Firmicutes to Bacteroidetes ratio. Thus, beneficial effects might be mediated by specific taxa. However, Metformin act in different fashion. In addition to its glucose-lowering and insulin sensitizing effects, metformin promotes SCFA-producing and mucin-degrading bacteria. Authors should briefly discuss how Metformin (and other antidiabetic drugs) could intefere with their results. Authors could also cite works by Montandon SA (Genes), Whang (Lancet) and Wu (Nature).
We would like to thank the reviewer for the valuable comments provided.
For the differential abundance analysis model, in the cross-sectional study, we decided that adjusting for the confounding variable “T2D prevalence” was enough to incorporate all the possible related interferences, included the use of metformin or other antidiabetic drugs.
What is important to mention is that no changes in metformin treatment have been reported in our participants after 12-month of follow-up, therefore we do not expect changes in gut microbiota caused by metformin use in the longitudinal analysis.
- It will be interesting to know the spectrum cultures in obese who lost weight after the intervention dependiing of antidiabetic drug administered.
We completelly agree with the reviewer that the spectrum cultures in obese individuals who lost weight after the intervention depending on the antidiabetic drugs administered, could be an intersring aspect to explore. Unfortunately, the sample size (just 19 individuals in the highest tertile of weight-loss and 26-28 individual in the second tertile of weight loss, under antidiabetic treatment) is not sufficient to successfully take over this type of analysis.
3.The microbiome spectrum in any study population will also depend on the regional prevalence. That information (comparing described results with other authors) is at least important as other comparisons.
We found the statement about the regional prevalence and possible effects on the gut microbiota totally appropriate, as the study of the core-microbiome from different areas is an important current topic, and it will be interesting to considered it for future studies.
As suggested by the reviewer, we have explored the percentage of participants belonging to the different recruiting centers in each tertile of baseline BMI or changes in body weight, and no significant differences were shown. Therefore, we do not expect differences due to regional prevalence in our final findings. However, we included the variable “recruiting center” in table 1 and table 2, in order to show that no differences in the percentage of individuals from each center were observed across different tertiles of BMI or changes in body weight.
Reviewer 3 Report
I would like to thank you for the opportunity to review this article and the authors for the work they have put in preparing the manuscript. This paper addresses an important issue as many researchers focus on the association between the gut microbiota and obesity.
The authors presented reasonable doubts about the results, such as geographical area, age, homogeneity of the group, and - most importantly in this case - the inability to establish causality properly; these are important issues when investigating the gut microbiota. Although the results are interesting, the question should be asked how the results could be useful in general practice, and few sentences could be provided (as a suggestion).
As a part of the PREDIMED-Plus clinical trial, the intervention group was encouraged to follow the Mediterranean diet; however, I wonder if the adherence to the nutritional recommendations was examined, as it could also affect the results.
Minor comment: please change "et el." to "et al." (line 288).
Author Response
I would like to thank you for the opportunity to review this article and the authors for the work they have put in preparing the manuscript. This paper addresses an important issue as many researchers focus on the association between the gut microbiota and obesity.
The authors presented reasonable doubts about the results, such as geographical area, age, homogeneity of the group, and - most importantly in this case - the inability to establish causality properly; these are important issues when investigating the gut microbiota. Although the results are interesting, the question should be asked how the results could be useful in general practice, and few sentences could be provided (as a suggestion).
As suggested, we provided few sentences to cover this question (lines 353-355).
As a part of the PREDIMED-Plus clinical trial, the intervention group was encouraged to follow the Mediterranean diet; however, I wonder if the adherence to the nutritional recommendations was examined, as it could also affect the results.
As suggested by the reviewer, we considered the variable “adherence to the Mediterranean diet”, determined by a 17-item questionnaire, where the score 0 means absence of adherence and the score 17 means maximum adherence. As no significant differences in the score were observed between tertiles of baseline BMI, we do not expect this variable to have significant effects in our final findings. In the longitudinal analysis, we stratified the population by tertiles of changes in body weight. This setup caused a disproportion in the distribution of individuals belonging to the intervention group between different tertiles (82.6% in T1; 54.8% in T2; 13 % in T3), and this is consequently reflected in significant differences in the adherence to the MedDiet after 12-month of follow-up, resulted higher in those tertiles containing more individuals belonging to the intervention group (6.7 ± 2.7 in T1; 4.4 ± 3.1 in T2; 2.2 ± 2.7 in T3). For this reason, we considered that adjusting the differential abundant analysis model for the variable “intervention group”, was enough to incorporate also the potential effects associated to the adherence to Mediterranean diet. In any case this outcome it has been explored more in detail in other studies from our group, for this reason we decided to do not deepen to do not overlap information between different studies.